# Association of Cesarean Delivery with Trajectories of Growth and Body Composition in Preschool Children

**DOI:** 10.3390/nu14091806

**Published:** 2022-04-26

**Authors:** Zijun Liao, Jing Wang, Fangfang Chen, Yiren Chen, Ting Zhang, Gongshu Liu, Xianghui Xie, Jun Tai

**Affiliations:** 1Capital Institute of Pediatrics, Beijing 100020, China; zijun_liao@yeah.net (Z.L.); yirenchen203@163.com (Y.C.); zhangtingcv@126.com (T.Z.); 2Tianjin Women’s and Children’s Health Center, Tianjin 300070, China; wangjing8162012@163.com (J.W.); liugongshu727@163.com (G.L.)

**Keywords:** cesarean delivery, obesity, growth, body composition, children, cohort

## Abstract

This study aimed to examine the association of cesarean delivery with trajectories of growth and body composition in preschool children. This ambidirectional cohort study was conducted between 2017 and 2020 in China. Information on the delivery mode, weight, and length/height of the children measured at routine healthcare visits was obtained from maternal and child health records. For three years while in kindergarten, children’s body mass index (BMI), fat mass index (FMI), fat-free mass index (FFMI), and percentage of body fat (FM%) were repeatedly measured. A BMI z score (zBMI) was calculated and standardized to WHO measures, and overweight and obesity were defined using the WHO reference. After adjustment for maternal age, maternal education, annual family income, prepregnancy BMI, gestational weight gain, gravidity, parity, gestational age, child sex, birthweight, breastfeeding duration, and the parent-reported dietary intake of the children, children born via cesarean delivery (*n* = 1992) versus those born vaginally (*n* = 1578) had higher zBMI growth rates beyond 36 months (*β*: 0.003; 95% CI: 0.001, 0.005 SD units/month) and elevated levels of FMI (*β*: 0.097; 95% CI: 0.026, 0.168 kg/m^2^), FM% (*β*: 0.402; 95% CI: 0.058, 0.745%) and zBMI (*β*: 0.073; 95% CI: 0.012, 0.133 units), but not FFMI (*β*: 0.022; 95% CI: −0.022, 0.066 kg/m^2^). The adjusted OR of overweight and obesity was 1.21 (95% CI: 1.04, 1.40). Cesarean delivery likely elevated zBMI growth rates and increased the risk of overweight and obesity in preschool children, with the elevation of fat mass but not fat-free mass.

## 1. Introduction

Cesarean delivery is one of the most common procedures as a life-saving intervention when indicated [1]. Cesarean rates have risen in recent decades worldwide, exceeding 30% in many middle- and high-income countries, such as the United States, Australia, Brazil, and China [2,3,4,5]. In the context of significant increases in nonmedically indicated cesareans, the increased health risks of mothers and children resulting from this procedure should be weighed [1,2]. A tremendous amount of literature suggests that cesarean delivery leads to short- and long-term health effects for children [1,6,7].

Childhood overweight and obesity has been an epidemic; an estimated 5.7 percent or 38.9 million children under the age of 5 around the world were overweight in 2020, with a rapid increase of nearly 6 million since 2000 [8]. Childhood overweight and obesity were also reported as possible adverse outcomes of cesarean delivery [9]. The plausible mechanisms included: decreased gut microbiota diversity, a lack of stress response, an altered immune phenotype, and an affected initiation and maintenance of breastfeeding [6,10,11,12]. Although mounting studies have examined the association between cesarean delivery and childhood obesity, the results have been inconclusive [9].

Prior studies examined the outcomes during a limited time, most of which used a single cross-sectional measure, and few examined the trajectory of child body mass growth. Additionally, most studies assessed obesity by solely relying on body mass index (BMI) instead of direct measures such as indicators of body fat. [13]. In this study, based on repeated measurement data, we not only explored the effects of cesarean delivery on childhood BMI as well as overweight and obesity but also included direct estimates of body composition measurements as outcomes to comprehensively reveal the impacts of cesarean delivery on children’s growth. In addition, as cesarean delivery and childhood obesity continue to rise, their interrelationship may change; children in the majority of these cohorts were born a decade ago [7], and the results may not reflect the effect of modern practice.

To address these gaps in knowledge, we conducted an ambidirectional cohort study to explore the association of cesarean delivery with trajectories of growth and body composition in children.

## 2. Materials and Methods

### 2.1. Participant’s Enrollment

The ambidirectional cohort study was implemented by Tianjin Women’s and Children’s Health Center and Capital Institute of Pediatrics in Tianjin municipality, China, from 2017 to 2020. From all of Tianjin’s 16 districts, 11 were selected using the stratified cluster sampling method, and 42 kindergartens were subsequently selected. Children in their 1st year of kindergarten (age 3 years) were recruited into the cohort study and were further followed up in their 2nd and 3rd years. Meanwhile, the data from before the children were recruited into the cohort were extracted from maternal and child health (MCH) records, which covered information on routine healthcare visits from the mothers’ pregnancy until the children reached 6 years of age. As the National Health Commission of China required, child healthcare visits were usually conducted at ages 1, 3, 6, 8, 12, 18, 24, and 30 months, as well as at ages 3 to 6 years.

In our cohort, children were excluded if they (1) were unable to give informed consent; (2) had been diagnosed with any condition or chronic disease, or used any drug known to affect growth and development; (3) had any acute diseases that prohibited the children from participating in a physical examination; (4) were part of a multiple birth; or (5) were without maternal or children’s data in MCH records.

### 2.2. Exposure and Covariates

We extracted data on the delivery mode from MCH records. The originally reported modes of delivery were classified into three categories: spontaneous vaginal delivery, assisted vaginal delivery, and cesarean delivery. In the present study, children born vaginally were included both via spontaneous and assisted vaginal delivery. We also extracted the potential confounders, including maternal age, maternal education, annual family income, maternal height, prepregnancy weight, weight before delivery, gravidity, parity, gestational age, child sex, birthweight, and breastfeeding duration. When children were in their 2nd year of kindergarten (age 4 years), their dietary intake, which was estimated by the usual amounts of consumed food, was reported by parents through a questionnaire, with the response options given on a 5-point response scale from low to high. Prepregnancy weight was self-reported at the mother’s first healthcare visit during the 1st trimester of pregnancy. Gestational weight gain was calculated as weight before delivery minus prepregnancy weight. Prepregnancy BMI was calculated as prepregnancy weight in kilograms divided by square of height in meters.

### 2.3. Outcome Measures

Throughout their three years in kindergarten, the height of the children was measured to the nearest tenth of a centimeter without shoes by trained staff with wall-mounted stadiometers according to the standardized protocol [14,15]. The children’s body weight, fat mass, fat-free mass, and percentage of body fat (FM%) were measured through a bioelectrical impedance analysis (BIA) with the body composition analyzer (Seehigher BAS-H, Beijing, China), which measures the impedances of the body and each limb in the standing position and is suitable for children aged 3 years and older, at frequencies of 1 kHz, 5 kHz, 50 kHz, 250 kHz, 500 kHz, and 1 MHz. The children were required to be fasting and to have an empty bladder. When being measured, the children stood on the platform without shoes and wearing light clothing and held both hands at a 45-degree angle away from the body; four tactile electrodes were in contact with the palm and thumb of both hands, and another four were in contact with the anterior and posterior aspects of the soles of both feet [14]. Additionally, to capture repeated measures of growth, we extracted the data of weight and length or height of children measured using standardized procedures at their routine healthcare visits from age 1 month to 6 years from MCH records. Body mass index (BMI), fat mass index (FMI), and fat-free mass index (FFMI) were calculated as weight, fat mass, or fat-free mass in kilograms divided by square of height/length in meters, respectively. The BMI z score (zBMI) was calculated and age- and sex-standardized to WHO measures by using the igrowup (<61 month) and AnthroPlus (≥61 month) packages for SAS on the basis of the WHO growth standards [16,17,18]. According to the WHO reference, children (0~5 years) above +2 SD were described as overweight and above +3 as obese, and children (5~19 years) above +1 SD and +2 SD were described as overweight and obese, respectively [17,18,19]; in our study, we used the term overweight and obesity to include both overweight and obesity in children (from 1 month to 6 years).

### 2.4. Statistical Analyses

We used the mean (standard deviation (SD)) or median (interquartile range (IQR)) to present continuous variables and frequencies (%) to present categorical variables. We explored the differences between the groups of the delivery mode using the *t* test for means, Kruskal–Wallis test for medians, and chi-square test for frequencies.

We estimated the zBMI growth trajectories in children from 0 to 6 years of age using piecewise linear mixed models, which model the trajectory with several linear splines and assess the linear associations within different periods rather than an entire trajectory [20,21,22]. A loess curve was used to determine the approximate age in months at which the direction or slope of growth changed (Appendix A), and to coincide with regular child healthcare visits [20,22], the knot points were fit at 6 and 36 months of age. The delivery mode was analyzed as the principal fixed effect to explore the association of cesarean section with zBMI growth trajectories, and then, multivariate models were further conducted by adjusting the maternal age at delivery (<35 or ≥35 years), maternal education (high school or less, college, or above college), annual family income (<10,000, 10,000~20,000, or ≥20,000 RMB), prepregnancy BMI (<18.5, 18.5∼23, 23∼27.5, or ≥27.5 kg/m^2^) [23], gestational weight gain (inadequate, appropriate, or excessive) [24], gravidity (1 or ≥2), parity (1 or ≥2), gestational age (preterm or full-term), child sex (male or female), birthweight (<2500, 2500∼3999, or ≥4000 g), breastfeeding duration (<6 or ≥6 months), and parent-reported dietary intake of children (high, relatively high, average, relatively low, or low). When considering the repeated measures collected, to take into account within-subject correlation, we used linear generalized estimating equation (GEE) models to explore the association of cesarean delivery with anthropometric and body composition measures (zBMI, FMI, FFMI, and FM%) in children (3 to 6 years), and used logistic GEE models for the outcome of overweight and obesity in children (1 month to 6 years); multivariate models reprised the adjustments of the same covariates as in the piecewise linear mixed models. We coded missing values of a covariate as a category with multivariate models. Subgroup analyses were performed to examine the coherence of associations by child birthweight (<2500, 2500~3999, or ≥4000 g) and child sex (male or female); interaction tests were performed by introducing a cross-product term in the multivariable-adjusted models.

A 2-tailed *p* value was considered significant at <0.05. Statistical analyses were performed in Stata 15.0, SAS 9.4 (SAS Institute Inc., Cary, NC, USA), and R software program version 4.0 (R Foundation for Statistical Computing, Vienna, Austria).

## 3. Results

Among the 3822 children that were recruited in their first year of kindergarten with measurements of body mass and composition, 252 were excluded because of missing data regarding delivery mode (*n* = 178) and being part of a multiple birth (*n* = 74). Finally, 3570 children remained in the analyses; 1578 (44.2%) children were born via vaginal delivery, and 1992 (55.8%) were born via cesarean delivery. Maternal and child characteristics varied by delivery mode (Table 1). Compared with mothers delivering a baby via vaginal delivery, those who delivered via cesarean were more likely to be older (28.5 years vs. 29.4 years), to have a lower proportion of above college (12.0% vs. 9.5%), to have a higher prepregnancy BMI (21.7 kg/m^2^ vs. 22.9 kg/m^2^), to have a higher proportion with excessive gestational weight gain (21.1% vs. 29.6%), to have a higher proportion of multiparous mothers (23.8% vs. 27.2%), to have a shorter gestational age (39.1 weeks vs. 38.9 weeks), to have a higher proportion of those delivering children weighing ≥4000 g (4.3% vs. 12.6%), and to have a lower proportion breastfeeding for a duration of ≥6 months (68.6% vs. 60.1%).

Children’s growth rates were rapid until 6 months of age (mean: 0.029; 95% CI: 0.019, 0.039 zBMI SD units/month), with a deceleration between 6 and 36 months (mean: −0.016; 95% CI: −0.017, −0.015 zBMI SD units/month), followed by a steady period after 36 months (mean: 0.0004; 95% CI: −0.001, 0.002 zBMI SD units/month); after adjustment of the covariates, the growth rates remained similar (Appendix A). Figure 1 shows the predicted zBMI growth trajectories from adjusted analyses by delivery mode. Although the zBMI growth trajectories were of the same trend in the two groups, children born via cesarean and vaginal delivery had different growth rates. Table 2 shows the differences in growth rates of zBMI by delivery mode in each growth period. In crude analyses, the growth rates were significantly different after the age of 36 months between children born via cesarean delivery and those born via vaginal delivery, with a higher growth rate (mean difference: 0.003; 95% CI: 0.0004, 0.005 SD units/month; *p* = 0.018) in the cesarean group. There was no significant difference in growth rates between groups at ages 0~6 months or 6~36 months (*p* > 0.05). After adjustment for confounders, the difference in growth rate between the two groups remained almost unchanged, with the difference in growth rate beyond 36 months being 0.003 SD units/month (95% CI: 0.001, 0.005 SD units/month; *p* = 0.014).

An association was evident between cesarean delivery and overweight and obesity in children (1 month to 6 years) in the crude analysis (OR: 1.44; 95% CI: 1.24, 1.66; *p* < 0.001). After multivariate adjustment, significantly increased odds remained (adjusted OR: 1.21; 95% CI: 1.04, 1.40; *p* = 0.011). Table 3 shows the differences in anthropometric and body composition measurements between kindergarten children (3 to 6 years) born via cesarean delivery and those born vaginally. After adjustments of covariates, children in the cesarean group had FMIs that were 0.097 kg/m^2^ (95% CI: 0.026, 0.168 kg/m^2^), FM% that were 0.402% (95% CI: 0.058, 0.745%), and zBMIs that were 0.073 units (95% CI: 0.012, 0.133 units) higher than those in the vaginal delivery group, whereas the FFMIs were similar between the two groups (*β*: 0.022, 95% CI: −0.022, 0.066 kg/m^2^). Omnibus tests of the interaction of delivery mode with children’s birthweight and sex fell short of statistical significance (*p* > 0.05, Appendix A).

## 4. Discussion

In this cohort study, we assessed the association between delivery mode and trajectories of growth and body composition in children. The zBMI growth rates were higher in children who were born via cesarean delivery beyond 36 months of age. The odds of overweight and obesity were approximately 21% higher in children born via cesarean delivery. In addition, cesarean delivery, compared with vaginal delivery, was associated with higher levels of FMI, FM%, and zBMI in kindergarten children (3 to 6 years), but not FFMI. 

### 4.1. Interpretation in Light of Other Studies

In prior systematic reviews and meta-analyses, obesity in children was analyzed as a dichotomous outcome, and the OR of the association ranged from 1.22 to 1.34 [7,9,25]. In accordance with the excess in the previous meta-analyses, our results showed that cesarean delivery could lead to a higher risk of childhood overweight and obesity (adjusted OR: 1.21; 95% CI: 1.04, 1.40) by repeated measures of the outcome of overweight and obesity from 1 month to 6 years in children. To the best of our knowledge, we did not find studies that investigated the association between delivery mode and children’s zBMI trajectories. In our study, the difference in the zBMI growth rates observed beyond 36 months of age persisted, indicating that cesarean delivery might be linked to an accelerated zBMI growth rate in later childhood.

In studies that estimated the association between cesarean delivery and childhood obesity, body fat measures were seldom explored. A Brazilian study found a null association of cesarean section with zBMI and FMI in 6-year-old children after covariate adjustment [13]. The prior study used dual-energy X-ray absorptiometry (DXA), a different method from our research, so the results might not be fully comparable. After measuring the children’s anthropometric and body composition index at 3 to 6 years, we found that cesarean delivery led to a higher level of FMI, FM%, and zBMI, but not FFMI. We speculated that cesarean delivery mainly contributed to the increase in fat mass rather than fat-free mass in children. Despite the significant effect of cesarean delivery on fat mass, the difference was relatively small between the two groups and should be tracked into later childhood. We did not find any evidence that the association between delivery mode and anthropometric and body composition measures was modified by child sex and birthweight, similar to other studies that did not find sex- or birthweight-specific growth patterns by delivery mode [9,25,26].

### 4.2. Possible Explanations

In many studies, the hygiene hypothesis was thought to be the underlying mechanism of cesarean and childhood obesity association. Compared with newborns born vaginally, those delivered via cesarean section are colonized by a gut microbiome resembling that of the mother’s skin and external environment rather than the maternal vaginal microbiota; the disordered gut microbiota composition contributes to the cesarean-delivered children’s development of obesity by increasing the capacity for energy harvest [10,27,28,29,30]. The altered feeding practices, hospitalization, and antibiotic use in the cesarean group also contributed to gut flora changes in cesarean-delivered children [27,30]. An altered fetal gene expression, metabolism, and hormonal responses in cesarean-delivered children because of nonexposure to labor are also reported as plausible causal factors [11,31].

### 4.3. Strengths and Limitations

This study has strengths. First, instead of solely using BMI, an indicator with some limitations [13], to assess obesity, we investigated a spectrum of indicators of body composition to comprehensively estimate the effects of cesarean delivery on childhood obesity. Second, unlike a single-point survey, we used repeated measurement data and examined the effect of cesarean delivery on zBMI growth rates during children’s specific growth periods. We also adjusted for key potential confounders simultaneously [9,13,32,33], such as annual family income, prepregnancy weight, and breastfeeding. Considering the temporal differences in obstetric practice [7] and other factors influencing children’s growth, our contemporary cohort could contribute to reflecting today’s epidemiology.

Our study also has limitations. First, the prepregnancy weight in our study was self-reported at the first visit during the first trimester, which was less accurate than the measured weight and resulted in a residual bias. Nonetheless, a systematic review showed that reporting error did not largely bias associations between pregnancy-related weight and birth outcomes [34]. Second, the children’s body composition indicators were measured using BIA rather than the gold standard DXA in this cohort study. The BIA method agreed well with the body composition assessment obtained with DXA [35,36,37,38,39], but it was likely to underestimate fat mass and FM% [36,37,38], with a similar trend seen in children stratified by age, sex, and BMI [37,38]. We used the BIA in the same ethnic group and the same age, with the same device, so the measurement bias was limited. Our results showed that cesarean delivery was associated with higher levels of fat mass and FM%. Therefore, if it is true that BIA systematically underestimated fat mass and FM% when compared with DXA, the significant differences in fat mass and FM% between the two groups by delivery mode also existed in our study, indicating that the ability to detect the true association seems unlikely to be influenced. Third, the study did not distinguish between elective and emergency cesarean delivery, perhaps limiting further exploration of the causal association, considering the conflicting results of associations with the two subgroups of cesarean and childhood obesity [7,40,41]. Additionally, objectively quantitated dietary intake and antibiotic use were not measured, and data on breastfeeding duration were collected without distinguishing between partial and exclusive breastfeeding, which resulted in a difference in children’s protein intake, likely leading to residual bias [9,42]. In our stratified analyses, the small sample size in some strata might lead to the limited power and greater instability of the results.

## 5. Conclusions

This contemporary ambidirectional cohort study found accelerated zBMI growth rates beyond 36 months of age and an increased risk of overweight and obesity in children born via cesarean delivery; cesarean delivery mainly contributed to elevated fat mass but not fat-free mass. The adverse effect of cesarean delivery should be weighed when the delivery mode is decided. Future studies are warranted to differentiate whether the effect of cesarean section on body composition was induced by the cesarean operation per se or the related medical indications.

## Figures and Tables

**Figure 1 nutrients-14-01806-f001:**
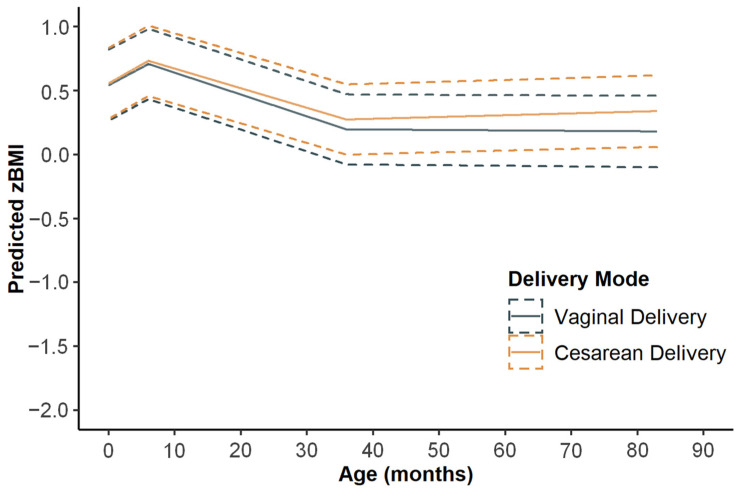
Predicted zBMI growth trajectories (95% CIs) from adjusted analyses by delivery mode. Adjusted for maternal age (<35 years), maternal education (high school or less), annual family income (<10,000 RMB), prepregnancy BMI (<18.5 kg/m^2^), gestational weight gain (inadequate), gravidity (1), parity (1), gestational age (preterm), child sex (male), birthweight (<2500 g), breastfeeding duration (<6 months), and parent-reported dietary intake of children (high). Abbreviations: zBMI, BMI z score.

**Table 1 nutrients-14-01806-t001:** Maternal and child characteristics according to delivery mode ^a^.

	Overall (*n* = 3570)	Vaginal Delivery (*n* = 1578)	Cesarean Delivery (*n* = 1992)	*p*
Maternal age at delivery (year), mean ± SD	29 ± 3.4	28.5 ± 3	29.4 ± 3.6	<0.001
Maternal age at delivery (year), *n* (%)				
<35	3301 (93.2)	1508 (96.4)	1793 (90.6)	<0.001
≥35	242 (6.8)	56 (3.6)	186 (9.4)	
Maternal education, *n* (%)				
Above college	378 (10.6)	188 (12.0)	188 (9.5)	0.034
College	2757 (77.6)	1205 (77.0)	1545 (78.1)	
High School or less	420 (11.8)	172 (11.0)	245 (12.4)	
Annual family income (RMB), *n* (%)				
<10,000	796 (23)	322 (21.1)	474 (24.5)	0.055
10,000–20,000	1651 (47.7)	747 (48.8)	904 (46.7)	
≥20,000	1017 (29.4)	461 (30.1)	556 (28.8)	
Prepregnancy BMI (kg/m^2^), mean ± SD	22.4 ± 3.5	21.7 ± 3.1	22.9 ± 3.7	<0.001
Prepregnancy BMI (kg/m^2^), *n* (%)				
<18.5	354 (10.5)	184 (12.4)	170 (9)	<0.001
18~23	1772 (52.6)	865 (58.2)	907 (48.2)	
23~27.5	943 (28)	355 (23.9)	588 (31.2)	
≥27.5	300 (8.9)	82 (5.5)	218 (11.6)	
Gestational weight gain, *n* (%)				
Inadequate	1156 (34.3)	575 (38.7)	581 (30.9)	<0.001
Appropriate	1341 (39.8)	597 (40.2)	744 (39.5)	
Excessive	872 (25.9)	314 (21.1)	558 (29.6)	
Gravidity, *n* (%)				
1	2174 (64.5)	1065 (67.5)	1130 (56.7)	<0.001
≥2	1199 (35.6)	513 (32.5)	862 (43.3)	
Parity, *n* (%)				
1	2653 (74.3)	1203 (76.2)	1450 (72.8)	0.019
≥2	917 (25.7)	375 (23.8)	542 (27.2)	
Gestational age (weeks), mean ± SD	39 ± 1.4	39.1 ± 1.5	38.9 ± 1.3	0.002
Gestational age, *n* (%)				
Preterm	133 (3.7)	68 (4.3)	65 (3.3)	0.109
Full-term	3437 (96.3)	1510 (95.7)	1927 (96.7)	
Child sex, *n* (%)				
Male	1853 (51.9)	804 (51)	1049 (52.7)	0.302
Female	1716 (48.1)	774 (49.1)	942 (47.3)	
Birthweight (g), mean ± SD	3380.7 ± 457.4	3306.1 ± 420	3439.7 ± 476.8	<0.001
Birthweight (g), *n* (%)				
<2500	86 (2.4)	46 (2.9)	40 (2)	<0.001
2500~3999	3165 (88.7)	1464 (92.8)	1701 (85.4)	
≥4000	319 (8.9)	68 (4.3)	251 (12.6)	
Breastfeeding duration (month), *n* (%)				
<6	1269 (36.1)	487 (31.4)	782 (39.9)	<0.001
≥6	2246 (63.9)	1066 (68.6)	1180 (60.1)	

^a^ The percentages of cases with missing data on maternal age, education, annual family income, prepregnancy BMI, gestational weight gain, gravidity, and breastfeeding duration were 0.8%, 0.4%, 3.0%, 5.6%, 5.6%, 5.5%, and 1.5%, respectively.

**Table 2 nutrients-14-01806-t002:** Mean differences in zBMI growth rates (SD units per month) for cesarean delivery compared with vaginal delivery (reference) during each growth period ^a^.

Growth Period	Adjusted Mean Differences (95%CI) ^b^	*p*	Unadjusted Mean Differences (95%CI)	*p*
0~6, month	−0.003 (−0.024, 0.017)	0.758	−0.003 (−0.024, 0.017)	0.755
6~36, month	0.001 (−0.001, 0.004)	0.260	0.001 (−0.001, 0.004)	0.272
>36, month	0.003 (0.001, 0.005)	0.014	0.003 (0.0004, 0.005)	0.018

^a^ Piecewise linear mixed models were used to examine mean differences in zBMI growth rates (SD units per month) by delivery mode (vaginal delivery group as the reference); ^b^ adjusted for maternal age, maternal education, annual family income, prepregnancy BMI, gestational weight gain, gravidity, parity, gestational age, child sex, birthweight, breastfeeding duration, and parent-reported dietary intake of children. Abbreviation: zBMI, BMI z score.

**Table 3 nutrients-14-01806-t003:** Associations of cesarean delivery with repeated anthropometric and body composition measures in kindergarten children (3–6 years) ^a^.

	*β* Coefficient	SE	95%CI	*p*
FMI, kg/m^2^				
Crude	0.194	0.039	(0.119, 0.270)	<0.001
Adjusted ^b^	0.097	0.036	(0.026, 0.168)	0.008
FFMI, kg/m^2^				
Crude	0.094	0.024	(0.048, 0.141)	<0.001
Adjusted ^b^	0.022	0.023	(−0.022, 0.066)	0.325
FM%				
Crude	0.821	0.184	(0.461, 1.181)	<0.001
Adjusted ^b^	0.402	0.175	(0.058, 0.745)	0.022
zBMI				
Crude	0.189	0.034	(0.123, 0.255)	<0.001
Adjusted ^b^	0.073	0.031	(0.012, 0.133)	0.018

^a^ Generalized estimating equations were used to examine the differences in repeated anthropometric and body composition measures by delivery mode (vaginal delivery group as the reference); ^b^ adjusted for maternal age, maternal education, annual family income, prepregnancy BMI, gestational weight gain, gravidity, parity, gestational age, child sex, birthweight, breastfeeding duration, and parent-reported dietary intake of children. Abbreviations: FMI, fat mass index; FFMI, fat-free mass index; FM%, percentage of body fat; zBMI, BMI z score.

## Data Availability

The data are not publicly available due to confidentiality reasons.

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
