# Peer review of "Association of Cesarean Delivery with Trajectories of Growth and Body Composition in Preschool Children"

_nutrients, 2022, doi:10.3390/nu14091806_

Round 1

Reviewer 1 Report

I read the new version of the paper with the correction of the reviewers and I have nothing to debate. 

Author Response

Thank you very much for the helpful comments.

Reviewer 2 Report

Abstract – include more detailed information on the potential confounders that were investigated.

Page 3 – line 99 - add reference to “according to the standardized protocol”

Page 3 – line 39 - parent-reported dietary intake of children (high, 139 relatively high, average, relatively low, low). Please describe in methodology at what age was this variable collect and how was the procedure. This information should not firs be mention on statistics chapter

Page 8 – line 260 – the subtitle “Implications”, is not clear what it means. The differences in the microbiota according to deliver have been previous well documented, but the title implications is excessive since there was no study of children microbioma. The differences in the microbioma could be one possible explanation for the exessive fat mass in caesarean delivery, but there are other possible explanations. If authors want to present this strong hypothesis in a separated item, the title could be “possible explanations”.

In spite the importance of the evaluation of breastfeed until 6 months is also very important the information of that breastfeeding was or not exclusive breastfeeding, since there is a difference in the protein intake a for partial vs exclusive breastfeeding (Asia Pac J Clin Nutr. 2018;27(6):1294-1301).  Suggest to add that in Strengths and limitations.

Author Response

Thank you very much for the helpful comments. We have addressed all comments and made changes accordingly. Please see the attachment.

This manuscript is a resubmission of an earlier submission. The following is a list of the peer review reports and author responses from that submission.

Round 1

Reviewer 1 Report

Overall it is good paper and I would like to give these comments to the authors of this paper: - They should indicate whether the values reported in the abstract of the paper was coefficient or means - The authors should interpret Figure. For example, - Why was it higher at 10 months then lower at 40 months and flat afterwards? - Table 3, was the estimates coefficient or means?

Author Response

Thank you very much for the helpful comments. We have addressed all comments and made changes accordingly. Please see the attachment.

Reviewer 2 Report

In this study, authors examined the association of cesarean delivery with trajectories of growth and body composition in preschool children.  Below my comments:

Dietary habits may have influenced the outcome. Table 1 shows that women who delivered by cesarean section had a higher pregestational BMI, a higher prevalence of overweight and obesity, and a higher weight gain during pregnancy than women who delivered by vaginal delivery, suggesting a consistently higher energy intake and less healthy dietary habits. Given that the food intake of such young children depends on parental choices, it seems likely that children born by cesarean section, as well as their mothers, had a higher energy intake. If so, the effect attributed to cesarean section would be smaller or not significant.

The effect on fat mass seems to be very small. Table 3 shows a difference of 0.4% in FM% changes, from 3 to 6 years, between children born by cesarean section and vaginal delivery. The magnitude of the differences observed should be discussed.

Please specify the model of bioimpedance analysis used. Is this a validated tool for such young children?

Author Response

(The authors gave the same response as above.)

Reviewer 3 Report

I congratulate the authors for this very well-designed study, with high novelty and good interest for the readers in this sector.

Here, I insert my personal comment and suggestion to improve the quality of your work.

  1. Line 73, please indicate 1st year of school or kindergarten. In this way is more understandable.
  2. Line 99, detail better how BIA analysis was performed.
  3. Line 157, there is a typo in the 1st bracket. 39.1 weeks are the gestational age of vaginal birth, not cesarean mode, like written in table 1.
  4. Please change figure 1 with the predicted zBMI growth trajectories from the adjusted analysis is more elegant, even if table 2 shows that there are no differences. 
  5. Line 216 "in prior systematic" insert word 'review'.
  6. Lines 227-229, precise that the FMI of the Brazilian study you cited used DXA for measurement of FMI, so the results are not totally comparable.

Author Response

(The authors gave the same response as above.)

Round 2

Reviewer 2 Report

I have no further comments for authors